# Maternal risk factors for underweight among children under-five in a resource limited setting: A community based case control study

Anil Sigdel[1]☉*, Hardik Sapkota[1]☉, Subash Thapa[2]☉, Anu Bista[3]‡, Anil Rana[1]‡

1 Department of Public Health, Chitwan Medical College, Bharatpur Chitwan, Nepal, 2 Research Unit of General Practice, University of Southern Denmark, Odense, Denmark, 3 Youth and Comprehensive Education, Family Planning Association of Nepal, Lalitpur, Nepal

☉ These authors contributed equally to this work.
‡ These authors also contributed equally to this work.
* sigdel.aanil@gmail.com

**Data Availability Statement:** All relevant data are within the manuscript and its supporting information files.

## Abstract

Previous studies conducted in Nepal have not identified the potential maternal risk for underweight among children under-five years of age in resource-poor settings. Therefore, to identify these risk factors for being underweight among children under-five years old, a community-based case-control study was conducted in a rural village in the Chitwan District in Nepal. Cases were defined as children who were diagnosed as underweight based on low weight per age, whereas controls were the children with normal weight for their age. Mothers of 93 cases and 186 controls were invited for an interview to collect the data. More than half of underweight children were female (51.6%) and nearly one third of them (31.2%) were aged 13–24 months. Nearly, 30% of the cases belonged to families in the lowest wealth quintile and 82% of cases were from food insecure families. Logistic regression analysis showed that children of mothers who were illiterate had 1.48 times the odds of being underweight compared to whose mothers were not illiterate (95% Confidence Interval [CI]: 1.53–3.07)). Children whose mother had not completed their postnatal care (PNC) were 3.16 more times likely to be underweight compared to children of mothers who completed PNC (95% CI: 1.24–8.03). The children who received care from other family members besides their mothers were 6.05 times more likely to be underweight (95% CI: 1.44–25.42); the children having mothers who had no income at all had 5.13 times the odds of being underweight (95% CI: 1.27–20.71) and children with diarrhea episodes within one month were 2.09 times more likely to be underweight (95% CI:1.02–4.31) compared to those children without any diarrhea episodes within one month. Women should be encouraged to take care of their children themselves, seek PNC services and take precautions to protect their children from diarrhea. Also, enabling factors such as education and improved income for women can help to reduce malnutrition among children.

**Funding:** The author(s) received no specific funding for this work.

**Competing interests:** The authors have declared that no competing interests exist.

## Introduction

Despite significant progress in the health status of children globally, under-nutrition among under children under-five remains an important public health problem in low-income countries. Undernutrition is the major cause of child mortality in low income countries. UNICEF estimates that malnutrition contributes to more than half of the nearly 12 million under-five deaths in developing countries each year, and 112 million children under-five years are underweight (<-2 weight-for-age z-scores) in low-income countries [1]. Malnutrition is a universal issue that no country in the world can overlook. About one-third of reproductive women are anemic, and each year about 20 million babies are born underweight [2]. Though significant steps are being taken to address malnutrition, there has been a slight decrease in being underweight since 2000, from 11.6% to 9.7% in 2016 [3]. Reduction of the prevalence of under-nutrition in under-five children is a top priority to reduce child mortality and morbidity [2]. Reduction in under-nutrition prevalence by 50% between 1990 and 2015 was among the most important targets of the first Millennium Development Goal (MDG). Nevertheless, progress remains slow, and most international goals set for improving child nutrition and health were not met by 2000 [2].

Multiple factors related to both mother and children contribute to malnutrition [1]. Most of all, it is the result of a combination of inadequate dietary intake and infection. Especially in low income settings, malnourished children are more likely to die as a result of common childhood diseases, to have lifetime disabilities and weakened immune systems, and to have poor education outcomes [1]. Moreover, maternal health status plays a significant role in determining the overall health condition of a child. For instance, children who were not feed colostrum is a proximal cause and mothers' level of education as more distal cause could lead to underweight among the children. The other major causes are maternal marital status, having a family size of five or above, children who were breast feed for less than 12 months, who had frequent diarrhea illness, and minimum dietary diversity [4,5].

Previous studies have highlighted several factors associated with the nutritional status of children. A study conducted in rural Nepal (Kamlaiya) shows that mother's age is another factor that affects the health of a child. Children who were born from mothers 24 years of age and over were found less likely to be under-weight than those with mothers below 24 years of age [6]. Additional factors (duration of breast feeding, family size, site of delivery, occupation of mother, and height of mothers) were also associated with the nutritional status of the child according to a study conducted in three districts of Nepal [7]. Other determinants of underweight include low birth weight; child's gender; illness, and household income [8]. These sorts of demographic characteristics are associated with underweight and are consistent in many other similar settings. A study conducted in Rwanda found that mothers above 35 years of age, with primary level or no education, and whose delivery was not assisted by a skilled service provider were more likely to have under-weight children [9]. Likewise, a study done in Ethiopia also found that age of child, illness, maternal decision-making power, maternal education, employment/occupation, and household income as independent and important interpreters of determining underweight children [10]. Despite the abundance of evidence in the field, little is known about the social and individual circumstances of women's life that affect the child's nutritional status, especially in a resource limited setting.

So far interventions to reduce underweight have mostly targeted affected children, such as growth monitoring, immunization, and control of infectious diseases. Interventions targeting women have not considered the impact that women's social and individual circumstances would have on the nutritional status of the children. For instance, maternal decision-making power is a strong factor of underweight children so interventions to improve mothers' role in

decision-making could contribute to alleviating the problem [10]. Likewise, the burden of being underweight can be significantly reduced by focusing an education intervention for the mothers on the importance of nutrition by community health workers [11]. Also, an emphasis on empowering women and improving the knowledge and practices of parents on appropriate infant and young child care practices plays a vital role in reducing childhood acute malnutrition [5]. Therefore, in this study, we aimed to identify the maternal risk factors for underweight among under 5 children in Padampur Village Development Committee (VDC) of Nepal.

## Materials and methods

A case-control study was conducted in rural Padampur VDC of Chitwan District Nepal. Padampur VDC was selected purposively because of its high prevalence (i.e. 37%) of underweight [1].

### Sampling and recruitment

Cases were underweight children (low weight for age). The criterion for underweight was: Z-value less than minus 2 standard deviations (SDs) below the median weight-for-age and the control were children not underweight, i.e. Z-value equal to or above minus 2 SDs. Cases and control were ascertained by using the weight-for-age indicator according to the World Health Organization's (WHO) new growth standards from 2006 for boys and girls [12]. A household survey had been conducted prior to the study to identify the cases and control in the selected VDC. The youngest child in the family was involved if there was more than one eligible child in one family. The cases were then selected by using systematic random sampling and each case was matched with two controls in term of place of residence. Only those mothers who have provided written consent for an interview and permit measuring the weight of their child were included in this study. Children with physical disability and children who were sick during data collection were not included.

The sample for the study was calculated using Epi Info 7 statcalc with the following values based on a study [13]: confidence level (CI) = 95%, power (1-β) = 80%, case control ratio = 1:2, percentage of control exposed = 10.5%, odds ratio = 2.86. The sample size was 258 i.e. Cases = 86 and controls = 172. Considering 10% non-response rate, the total sample for the study was 284. However, five interviews were incomplete and were not included in the analysis. Hence, there were 279 samples analyzed in the study i.e. 93 cases and 186 controls [Fig 1].

### Questionnaire and measurement

A pre-structured questionnaire was adopted from the Nepal Health Demographic Survey (NDHS) 2011[14] and other similar previous studies conducted in Nepal [1,4,6,8,15,16]. It was then back translated into the native language (Nepali). Pre-testing of the tool was conducted in a geographically similar VDC of Chitwan district, Jutpani to test for cultural acceptability and adaptability. A two-day training was conducted for Female Community Health Volunteers (FCHV) on interview methods, the data collection tool, ethical consideration and anthropometric measurement (i.e. weight of under-five children). Weight was measured using a calibrated Salter scale, for use up to 25 kg with 100 grams precision. The child's weight was taken without shoes and with light clothes. The weight of children less than 6 months was taken by an infant weighing scale (infant spring scale) that is used in community-based newborn care programs in Nepal [16]. Children's ages were verified either through the birth registration certificate or immunization card issued from the health facility.

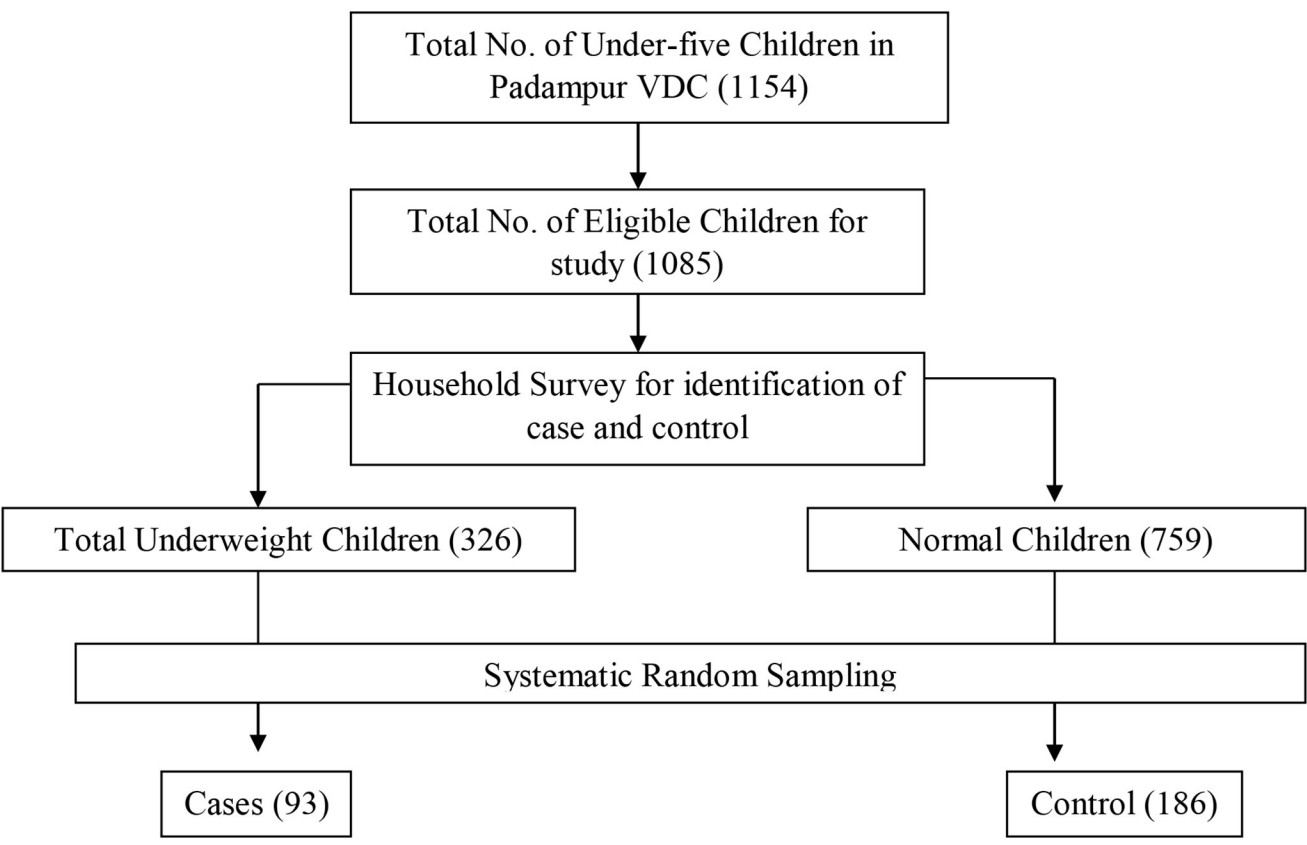

**Fig 1. Sampling procedure for selection of cases and control.**

### Measurement of variables

Underweight was the outcome variable of this study. Underweight was ascertained by using the weight-for-age indicator. The criterion was Z-value less than minus 2 standard deviations (SDs) below the median weight-for-age according to WHO new child growth standards 2006 [12].

Independent variables such as socio-demographic and economic factors (sex of children, ethnicity, mother's education, mother's earning status), Health service utilization factors (Antenatal Care [ANC] Visit, PNC Visit, place of delivery), Family related factors (birth order, age of mother at first child birth, number of children; height of mother; caregiver of children), history of childhood diseases (birth weight, occurrence of diarrhea, occurrence of Acute Respiratory Infection [ARI]), Feeding Practices (initiation of breast feeding, colostrum feeding, complementary feeding; exclusive breast feeding, feeding times per day, minimum food diversity, feeding during illness) were mainly reported by the child's mother during the interview. However, environmental and hygiene factors (water purification methods, availability of latrines and cooking fuel) were observed by interviewers on the day of interview. Household food security status was measured as secure versus insecure using the household food insecurity scale developed by the Food and Nutrition Technical Assistance (FANTA) [17]. The major independent and outcome variables are shown in Table 1.

### Data collection

At first, FCHVs approached the mothers of the selected children and asked for their consent to measure the weight of their children and to participate in an interview. Face to face interviews

**Table 1. Categories of variables under study.**

| SN | Variables | Categories of Variables |
|---|---|---|
| **Outcome Variables** | | |
| 1 | Nutritional status of under-5 children | Under-weight (low weight for age) |
| **Independent Variables** | | |
| **Socio-demographic Variables** | | |
| 2 | Education of mother | Illiterate; Literate |
| 3 | Sex of child | Male; Female |
| | Mother's earning status | Yes; No |
| **Family Related Variables** | | |
| 4 | Child's birth order | Second or less; Third or more |
| 5 | Height of mother | Less than 150 cm; 150 cm and more |
| 6 | Care giver of children | Mothers; Others |
| | Age of mother at first childbirth | Between 20–30 Years; Others |
| **Health Service Utilization Factors** | | |
| 7 | ANC visit | Less than 4 visits; 4 or more visit as per protocol (Visits during 4,6,8 and 9th months of pregnancy as suggested by Ministry of Health and Population (MoHP), Nepal) |
| 8 | Place of delivery | Health Facility; Home and other place |
| 9 | PNC visit within 45 days after delivery | Yes; No |
| **Environmental and Hygiene Related Factors** | | |
| 10 | Cooking fuel | Woods; LPG gas |
| 11 | Availability of toilet | Yes; No |
| 12 | Types of toilet | Dug well Latrine; Modern |
| 13 | Water purification methods | Yes; No |
| **Feeding Practices Variables** | | |
| 14 | Initiation of breast feeding | Within one hour; After one hour |
| 15 | Colostrum feeding | Yes; No |
| 16 | Exclusive breast feeding | 6 months or above; Less than 6 months |
| 17 | Feeding times per day | Less than 3 times; 3 times or more |
| 18 | Minimum food diversity | Standard (4 or more types of food out of 7 types (Grains, roots and tubers; Legumes and nuts; dairy products; Flesh foods; Eggs; Vitamin A rich fruits and vegetables and Other fruits and vegetables) consumed in a day; Below Standard (consumed less than 4 types of food) |
| 19 | Feeding during illness | Less than usual day; As usual or more |
| **History of Childhood diseases related variables** | | |
| 20 | Birth weight | Less than 2.5 kg; 2.5 Kg or more |
| 21 | Occurrence of ARI | No ARI; One or more times |
| 22 | Occurrence of diarrhea | No Diarrhea; One or more times |
| 23 | **Food Security Status** | Secure; Insecure |

took place in a separate room of the selected household by FCHVs using the semi-structured questionnaire. On average, it took about 45 minutes to complete one interview.

## Ethical considerations

The ethical approval was obtained from Institutional Review Committee (IRC) of Chitwan Medical College (CMC). Prior to data collection, permission from the District Public Health

Office (DPHO), Chitwan was obtained. Health workers and FCHVs from the selected VDC were briefed on the objective of the study. Written informed consent was also taken from mothers prior to interview. In the case of illiterate mothers, verbal consent was obtained from the responding mothers and written consent was obtained from other family members. Also, all mothers were explained about the objectives, benefits, risks and use of the study prior to the interview. No personal identifiers were collected during the interview. An alpha-numeric code was assigned to each respondent to maintain confidentiality. After completing this procedure, the interview was conducted, and the weight of children was measured.

### Data analysis

Data was analyzed using SPSS version 22 for windows. A chi-square test was used to compare the proportions between cases and controls. Logistic regression analysis was carried out to identify factors associated with being underweight. All the variables with a p-value of 0.20 in bi-variate analysis were entered in a multi-variate model. Forward stepwise logistic model was used with a p value of 0.20 for entry and 0.10 for exit. All polychotomous variables were dichotomized as seen in Table 1 for the logistic regression analysis. Hosmer and Lemeshow goodness of fit test was used to test the fitness of the model.

## Results

Approximately one-third of cases (31.2%) were in 13–24 months of age. The mean of age of cases and controls was 22.1 and 22.4 months respectively. Half of all cases were female (51.6%). Hindu children (69.9%), Disadvantaged Janajatis (caste/ethnic minority groups) (80.6%), households with food insecurity (82.8%) and lowest quintile households (29%) accounted for a higher proportion of cases as summarized in Table 2.

Forward stepwise logistic regression analysis was carried out with underweight as a dependent variable and nine independent variables with a p value less than 0.20 in bivariate analysis. The final model is summarized in Table 3. Children of mothers who were illiterate were 1.5 times more likely to be underweight than children of mothers who were literate (95% CI: 1.53–3.08). Children of mothers who did not received PNC services were 3.12 times more likely to be underweight than the children of mothers who visited PNC (95% CI: 1.24–8.03). Children who received care from other members of the family besides their mother were 6.04 times more likely to be underweight than control (95% CI: 1.44–25.42). Similarly, mothers who had no income were 5.13 times more likely to be underweight compared to those who had monthly income (95% CI: 1.27–20.71). Likewise, the episodes of diarrhea and underweight were found to be significantly associated. It was found that children who had one or more diarrhea episodes with in a one-month period were 2.09 times more likely to be underweight than those children who did not experience episodes of diarrhea (95% CI 1.02–4.31). Other factors like age at child birth, sex of children, ANC visit, breast feeding within one hour of delivery and exclusive breast feeding were not significantly associated with underweight in the multivariate analysis as shown in Table 3.

Children of mothers who have not received PNC services, children who received care from others besides their mother, mothers having income and children with one or more diarrhea episodes were statistically significant independent factors for underweight among children under 5 years of age. However, other factors like age at childbirth, sex of children, ANC services received by mothers, breast feeding within one hour of birth, exclusive feeding, colostrum feeding, and complementary feeding were not statistically significant risk factors associated with underweight. The p value for Hosmer and Lemeshow goodness of the fit was 0.85. The Nagelkerke R Square for the model was 0.38.

Table 2. Socio-demographic and economic characteristics of cases and control.

| Characteristics | Nutritional Status (weight for age) | | Total n = 279 (%) |
| --- | --- | --- | --- |
| | Case n = 93 (%) | Control n = 186 (%) | |
| **Age of children(month)** | | | |
| 0–12 | 28 (30.1) | 61 (32.8) | 89 (31.9) |
| 13–24 | 29 (31.2) | 44 (23.7) | 73 (26.2) |
| 25–36 | 19 (20.4) | 43 (23.1) | 62 (22.2) |
| 36–48 | 12 (12.9) | 23 (12.4) | 35 (12.5) |
| 49–60 | 5 (5.4) | 15 (8.1) | 20 (7.2) |
| Mean age (± SD) | 22.1 ± 14.9 | 22.4 ± 15.8 | 22.3 ± 15.5 |
| **Sex of the children** | | | |
| Male | 45 (48.4) | 115 (61.8) | 160 (57.3) |
| Female | 48 (51.6) | 71 (38.2) | 119 (42.7) |
| **Religion** | | | |
| Hindu | 65 (69.9) | 123 (66.1) | 188 (67.4) |
| Buddhist | 12 (12.9) | 37 (19.9) | 49 (17.6) |
| Christian | 16 (17.2) | 26 (14.0) | 42 (15.1) |
| **Ethnicity** | | | |
| Dalit | 11 (11.8) | 18 (9.7) | 29 (10.4) |
| Disadvantaged Janajatis | 75 (80.6) | 142 (76.3) | 217 (77.8) |
| Relatively advantaged Janajatis | 3 (3.2) | 7 (3.8) | 10 (3.6) |
| Upper caste groups | 4 (4.3) | 19 (10.2) | 23 (8.2) |
| **Educational status of the mother** | | | |
| Illiterate | 30 (32.3) | 30 (16.1) | 60 (21.5) |
| Informal education | 12 (12.9) | 19 (10.2) | 31 (11.1) |
| Primary education | 47 (50.5) | 89 (47.8) | 136 (48.7) |
| Secondary education | 3 (3.2) | 36 (19.4) | 39 (14.0) |
| Inter-mediate education (10+2) | 1 (1.1) | 9 (4.8) | 10 (3.6) |
| Higher education | 0 (0.0) | 3 (1.6) | 3 (1.1) |
| **Occupation of mother** | | | |
| Housewife | 87 (93.5) | 152 (81.7) | 239 (85.7) |
| Involved in income generating works | 6 (6.5) | 34 (18.3) | 21 (7.5) |
| **Wealth quintile** | | | |
| Low | 27 (29.0) | 28 (15.1) | 55 (19.6) |
| Second | 26 (28.0) | 30 (16.1) | 56 (20.1) |
| Middle | 19 (20.4) | 37 (19.9) | 56 (20.1) |
| Fourth | 12 (12.9) | 44 (23.7) | 56 (20.1) |
| Highest | 9 (9.7) | 47 (25.3) | 56 (20.1) |
| **Food security status** | | | |
| Insecure | 77 (82.8) | 108 (58.1) | 185 (66.3) |
| Secure | 16 (17.2) | 78 (41.9) | 94 (33.7) |

## Discussion

The study conducted in Zambia concluded that children with illiterate mothers were 1.12 times more likely to be underweight (95% CI: 1.11–1.12) [18]. Another study in Ethiopia also showed that children whose mothers have a basic education were 0.18 times less likely to be underweight than illiterate mother (95% CI: 0.08–0.41) [19]. All these findings resembled with this study, where the children of illiterate mothers were 1.5 times more likely to be underweight. Previous studies have shown that educated mothers are better informed about optimal

**Table 3. Factors Associated with underweight among under 5 children.**

| Variables | Case n (%) | Control n (%) | COR (95% CI) | AOR (95% CI) |
|---|---|---|---|---|
| **PNC visits within 45 days after delivery** | | | | |
| No (Less than 3) | 76 (81.7) | 111 (59.7) | 3.02 (1.65–5.51) | 3.31 (1.40–7.78) * |
| Yes (3 or More) | 17 (18.3) | 75 (40.3) | 1 | 1 |
| **Care giver of children** | | | | |
| Others | 15 (16.1) | 10 (5.4) | 3.38 (1.46–7.87) | 5.51 (1.48–20.54) * |
| Mother | 78 (83.9) | 176 (94.6) | 1 | 1 |
| **Income of mother** | | | | |
| No | 87 (93.5) | 148 (84.2) | 3.72 (1.51–9.16) | 4.29 (1.25–14.78) * |
| Yes | 6 (6.5) | 38 (20.4) | 1 | 1 |
| **Age at childbirth** | | | | |
| <20 and >30 years | 57 (61.3) | 75 (40.3) | 2.34 (1.41–3.90) * | 1.74 (0.84–3.57) |
| 20–30 years | 36 (38.7) | 111 (59.7) | 1 | 1 |
| **Sex of children** | | | | |
| Female | 48 (51.6) | 71 (38.2) | 1.73 (1.04–2.86) * | 0.62 (0.30–1.26) |
| Male | 45 (48.4) | 115 (61.8) | 1 | 1 |
| **Educational status of mothers** | | | | |
| Illiterate | 30 (32.3) | 30 (16.1) | 2.47 (1.38–4.44) * | 1.48 (1.53–3.07) * |
| Literate | 63 (67.7) | 156 (83.9) | 1 | 1 |
| **ANC Visits** | | | | |
| No (Less than 4) | 56 (72.7) | 86 (50.3) | 2.64 (1.47–4.73) * | 1.63 (0.74–3.59) |
| Yes (4 or More) | 21 (27.3) | 85 (49.7) | 1 | 1 |
| **Initiation of breast feeding** | | | | |
| After one hour | 48 (51.6) | 64 (34.4) | 2.03 (1.22–3.37) * | 1.35 (0.64–2.82) |
| Within one hour | 45 (48.4) | 122 (65.6) | 1 | 1 |
| **Exclusive breast feeding** | | | | |
| <6 and ≥6 Months | 59 (63.4) | 86 (46.2) | 2.02 (1.21–3.36) * | 1.53 (0.71–3.30) |
| Up to 6 months | 34 (36.6) | 100 (53.8) | 1 | 1 |
| **Episodes of Diarrhea within one month of survey** | | | | |
| One or more times | 48 (51.6) | 58 (31.2) | 2.35 (1.41–3.93) * | 2.09 (1.02–4.31) * |
| No diarrhea | 45 (48.4) | 128 (68.8) | 1 | 1 |

*statistically significant at $p < 0.05$, CI = Confidence Interval; COR = Crude Odds Ratio; AOR = Adjusted Odds Ratio; 1 = Reference

child care practices, have better practices in terms of hygiene, feeding and childcare during illness, have a greater ability to use the health system, are more empowered to make decisions and are more likely to have financial resources to care for and feed children [7].

Income status of mothers was a risk factor associated with underweight among children under-five years of age. This study showed that children whose mothers had no income were more than three times more likely to be underweight as compared to those mothers who have monthly income. This finding is aligned with the study conducted in Kailali district of Nepal which showed that risk factors for stunting comprised mothers without earning [15]. Likewise, a study conducted in Botswana showed that children of unemployed parents were more likely to be underweight [20]. These studies both show that nutritional status of children is highly associated with the earning status of mothers. A possible explanation could be that mothers who have income have better access to the health system and are more empowered to make decisions to care for and feed children.

Likewise, children who received care from other members of the family besides their mother were more than three times more likely to be underweight than those who were cared by their mother. These findings resembled the study conducted in Botswana, which showed that children raised by other members of the household were 5.67 times more likely to be underweight than raised by their own mothers (95% CI: 1.30–24.73) [20]. This might be because mothers are more connected and are more protective towards their children than other family members. Any minor deviation in health and hygiene habits of children are easily noticed by mother and take actions timely to prevent further complications.

PNC visits were highly associated with underweight in multivariate logistic regression analysis. Children whose mothers did not visit PNC were more than three times more likely to be underweight than the children of mothers visited PNC. The finding resembled the study conducted in Kunchha VDC of Nepal where children whose mothers did not visit were more than seven times more likely to be malnourished [21]. PNC is the period after the delivery of the child which includes routine clinical examination and observation of the woman and her child. Mothers visiting PNC could be more informed about their health and the health of their children and seek health services more frequently.

The occurrence of diarrhea within a one-month period is a risk factors associated with underweight among under five children. Children suffering from diarrhea one or more times during one-month period were two times more likely to be underweight than those who do not suffer from diarrhea. This finding aligns with the study conducted in Vietnam where it was found that children with diarrhea in the last two weeks are more than two times more likely to be underweight [8]. Likewise, other studies also revealed that child illness has a significant relationship with underweight of children [5,20]. This is clear because there is a reciprocal relationship with diarrhea leading to malnutrition while malnutrition predisposes to diarrhea. Infections play a major role in the etiology of under nutrition because they result in increased needs and high energy expenditure, lower appetite, nutrient losses due to vomiting, diarrhea, poor digestion, malabsorption of nutrients and disruption of metabolic equilibrium [10].

## Conclusions

In this study, we found that children whose mother has not completed her postnatal care, and who were illiterate, the children who received care from other family members besides their mothers, the children whose mothers had no income at all, and children with diarrhea episodes within one month were found to be at a significant risk of being underweight. It is recommended that women should be encouraged to take care of their children oneself, seek PNC services and take precautions to protect children from diarrhea. Also, increasing education and income generation for women can help to reduce malnutrition among children.

## Limitations

This study has some limitations. First, due to the retrospective nature of the study, it could be argued that the observed associations are due to recall bias. Second, the study was carried out in Padampur VDC of Nepal so the findings may have limited generalization. Third, the limited number of sample size may have led to a low statistical power to detect statistically significant differences. Forth, the use of self-reports might raise concerns about recall bias and social desirability tendencies. However, the robust study design and consistent results with previous studies conducted in Nepal and elsewhere suggest that the observed associations are valid.

## Supporting information

**S1 Appendix. Questionnaire for the survey.**
(DOCX)

**S1 File.**
(DOCX)

**S1 Data.**
(SAV)

## Acknowledgments

The authors would like to thank District Public Health Office (DPHO) Chitwan, Padampur Village Development Committee (VDC) and all Female Health Care Volunteers for their help and support during the field work of this study. We would like to extend our sincere gratitude towards Natalie Linton and Hannah Marqusee for professional editing of the manuscript.

## Author Contributions

**Conceptualization:** Anil Sigdel.

**Formal analysis:** Anil Sigdel, Hardik Sapkota.

**Investigation:** Hardik Sapkota, Anu Bista, Anil Rana.

**Methodology:** Anil Sigdel, Hardik Sapkota.

**Software:** Anil Sigdel, Hardik Sapkota.

**Writing – original draft:** Anil Sigdel, Subash Thapa, Anu Bista, Anil Rana.

**Writing – review & editing:** Anil Sigdel, Hardik Sapkota, Subash Thapa, Anu Bista, Anil Rana.

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
