## [Decision Letter · Decision Letter 0]

8 Apr 2020

PONE-D-20-02615

Maternal Risk Factors for Underweight among Under-five Children in a Resource Limited Setting: A Community Based Case Control Study

PLOS ONE

Dear Mr. Sigdel,

Thank you for submitting your manuscript to PLOS ONE. After careful consideration, we feel that it has merit but does not fully meet PLOS ONE’s publication criteria as it currently stands. Therefore, we invite you to submit a revised version of the manuscript that addresses the points raised during the review process.

We would appreciate receiving your revised manuscript by May 23 2020 11:59PM. To enhance the reproducibility of your results, we recommend that if applicable you deposit your laboratory protocols in protocols.io, where a protocol can be assigned its own identifier (DOI) such that it can be cited independently in the future. For instructions see: http://journals.plos.org/plosone/s/submission-guidelines#loc-laboratory-protocols

We look forward to receiving your revised manuscript.

Kind regards,

Vijayaprasad Gopichandran

Academic Editor

PLOS ONE

2. Please include additional information regarding the survey or questionnaire used in the study and ensure that you have provided sufficient details that others could replicate the analyses. For instance, if you developed a questionnaire as part of this study and it is not under a copyright more restrictive than CC-BY, please include a copy, in both the original language and English, as Supporting Information. Moreover, please include more details on how the questionnaire was translated.

Reviewers' comments:

Reviewer's Responses to Questions

**Comments to the Author**

1. Is the manuscript technically sound, and do the data support the conclusions?

Reviewer #1: Yes

2. Has the statistical analysis been performed appropriately and rigorously? 

Reviewer #1: Yes

3. Have the authors made all data underlying the findings in their manuscript fully available?

Reviewer #1: No

4. Is the manuscript presented in an intelligible fashion and written in standard English?

Reviewer #1: Yes

5. Review Comments to the Author

Reviewer #1: Detail comments are attached in PDF file in track mode for reference.

Author: 1) Scientific clarifications in methodology part 2) Presentation of result (Tables) can be improved..Specifically Table 3. 3) Minor essential language editings are required.

6. PLOS authors have the option to publish the peer review history of their article (what does this mean?). If published, this will include your full peer review and any attached files.

Reviewer #1: Yes: PRIYAMADHABA BEHERA

---

## [Author Response · Author response to Decision Letter 0]

24 Apr 2020

Response to Editors Comments

Authors Response: Changes made as per the PLOSOne format.

2. Please include additional information regarding the survey or questionnaire used in the study and ensure that you have provided sufficient details that others could replicate the analyses. For instance, if you developed a questionnaire as part of this study and it is not under a copyright more restrictive than CC-BY, please include a copy, in both the original language and English, as Supporting Information. Moreover, please include more details on how the questionnaire was translated.

Authors Response: Thanks for the suggestion. Questionnaire (both English and Nepali Version) for the survey have been uploaded as S2 Appendix as supporting information. A pre-structured questionnaire was adopted from the Nepal Health Demographic Survey (NDHS) 2011 and other similar previous studies conducted in Nepal. It was then back translated into the native language (Nepali). Pre-testing of the tool was conducted in a geographically similar VDC of Chitwan district, Jutpani to test for cultural acceptability and adaptability and questionnaire was finalized after making necessary changes based on the pre-test findings. 

 Authors Response: Updated as suggested. 

Authors Response: Updated as suggested. 

Response to Reviewer Comments

Reviewer Comment

Here authors objective to study chronic malnutrition or acute malnutrition 

Authors Response

Thank you for the comment. In this study, our objective was to study the acute form of malnutrition using community-based data. 

Reviewer Comment

Why you took weight for age? What you think if we have taken stunting (Yes/No) ...Will be it better? I thinks authors are interested in chronic malnutrition

Authors Response

In the present study, we used ‘weight-for-age’ to identify the children with underweight (acute malnutrition). We really appreciate your suggestion that investigating the factors for stunting, i.e., chronic malnutrition would also very relevant in the present context, and this is what we have planned to do as a future project that will analyze the Demographic and Health Survey data from 2001 to 2016.

Reviewer Comment

Percentage of control exposed 10.5%. Why you took this?

Authors Response

As it has been made clear in the manuscript that the percentage of control exposed, i.e., 10.5% was used to calculate the data using Epi Info statcalc. Please refer to the picture attached for the parameters that were used while calculating the sample size for this study that we have uploaded in response to reviewers file.

Reviewer Comment

What was the precision of the instrument? Near to half Kg?

Authors Response

The precision of the instrument was 100 grams, and this information has been mentioned in the ‘questionnaire and measurement’ sub-section of the revised manuscript. 

Reviewer Comment

How you weigh the older children? 4/5 years (who were walking). by salter (Maximum weight range for the use salter scale)

Authors Response

Thank you for the comment and as you have said, the weight of 4/5 years children were measured by salter scale. The maximum weight range for the use of salter scale is considered to be 25 kg and can be used up to children under 9. 

Reviewer Comment

Are the persons (FCHVs) who collected the data, were also involved in delivery of health care services (ANC, PNC). If Yes, there is possibility of interviewer bias in the study. This could be limited with independent data collector who were not involved with health care services delivery. 

Authors Response

Thank you for pointing this out. As the research team was also aware of the possibility of interviewer bias and this is why FCHVs, unlike other community health workers, who are not directly involved in delivering the health care services were involved to conduct the field work. FCHVs in Nepal work as health message communicators in the community, and as such, they have no direct influence on the population of interest in this study through their personal and professional relationship in relation to health services provision.

Reviewer Comment

Forward stepwise logistic regression model .....

Authors Response

The suggested change has been incorporated. 

Reviewer Comment

add SD (+-SD) in heading

Authors Response

The suggested change has been incorporated. 

Reviewer Comment

The table is incomplete. The variables also need to mention the number of children in each category. Heading ....two more column need to be added in table...One for case (n) another for control (n). 

Authors Response

Changes have been incorporated, as suggested. Two columns have been added in table 3, as suggested by you. 

Reviewer Comment

Is it during initial 6 months after delivery

Author Response

Thank you for the comment. This is within 45 days after delivery as per the protocol of Ministry of Health and Population. Timeframe has been updated on Table 1: Categories of Variables under study. 

Reviewer Comment

How many children are there ....with care givers other than 

Authors Response

Out of the total children, 9% were looked after other mothers. Values have been updated in Table 3 as suggested. 

Reviewer Comment

Duration which period ?

Authors Response

This is within one month of survey. Time frame for occurrence of diarrhea has been updated on Table 1: Categories of Variables under study. 

Reviewer Comment

"strong study design" is a vague word it needs to deleted

Authors Response

Deleted as suggested. We have replaced the phrase ‘strong study design’ with ‘robust study design’ in the ‘study limitations’ sub-section of the revised manuscript.

---

## [Editor Report · Decision Letter 1]

28 Apr 2020

Maternal Risk Factors for Underweight among Under-five Children in a Resource Limited Setting: A Community Based Case Control Study

PONE-D-20-02615R1

Dear Dr. Sigdel,

We are pleased to inform you that your manuscript has been judged scientifically suitable for publication and will be formally accepted for publication once it complies with all outstanding technical requirements.

With kind regards,

Vijayaprasad Gopichandran

Academic Editor

PLOS ONE
---

## [Editor Report · Acceptance letter]

12 May 2020

PONE-D-20-02615R1 

Maternal Risk Factors for Underweight among Children Under-five in a Resource Limited Setting: A Community Based Case Control Study 

Dear Dr. Sigdel:

I am pleased to inform you that your manuscript has been deemed suitable for publication in PLOS ONE. Congratulations! Your manuscript is now with our production department. 

With kind regards,

on behalf of

Dr. Vijayaprasad Gopichandran 

Academic Editor

PLOS ONE